# Prominent Follicular Keratosis in Multiple Intestinal Atresia with Combined Immune Deficiency Caused by a TTC7A Homozygous Mutation

**DOI:** 10.3390/genes13050821

**Published:** 2022-05-04

**Authors:** Andrea Diociaiuti, Roberta Caruso, Silvia Ricci, Rita De Vito, Luisa Strocchio, Daniele Castiglia, Giovanna Zambruno, May El Hachem

**Affiliations:** 1Dermatology Unit, Bambino Gesù Children’s Hospital, IRCCS, Piazza Sant’Onofrio 4, 00165 Rome, Italy; may.elhachem@opbg.net; 2Genodermatosis Unit, Genetics and Rare Diseases Research Division, Bambino Gesù Children’s Hospital, IRCCS, Piazza Sant’Onofrio 4, 00165 Rome, Italy; giovanna.zambruno@opbg.net; 3Department of Pediatric Oncohematology, Cell and Gene Therapy, Bambino Gesù Children’s Hospital, IRCCS, Piazza Sant’Onofrio 4, 00165 Rome, Italy; roberta.caruso@opbg.net (R.C.); luisa.strocchio@opbg.net (L.S.); 4Immunology Unit, Department of Health Sciences, Meyer Children’s University Hospital, Viale Pieraccini 24, 50139 Florence, Italy; silvia.ricci@meyer.it; 5Pathology Unit, Bambino Gesù Children’s Hospital, IRCCS, Piazza Sant’Onofrio 4, 00165 Rome, Italy; rita.devito@opbg.net; 6Laboratory of Molecular and Cell Biology, IDI, IRCCS, Via Monti di Creta 104, 00167 Rome, Italy; d.castiglia@idi.it

**Keywords:** dermoscopy, ichthyosis, keratinocyte apoptosis, TTC7A mutation

## Abstract

Multiple intestinal atresia with combined immune deficiency (MIA-CID) is an autosomal recessive syndrome due to mutations in the TTC7A gene implicated in the polarization of intestinal and thymic epithelial cells. MIA-CID is lethal in the first year of life in the majority of patients. Dermatological manifestations have been reported in a few cases. We describe a child affected with MIA-CID due to a previously unreported TTC7A homozygous missense mutation. Surgery for bowel occlusion was performed in the first days of life. The patient was totally dependent on parenteral nutrition since birth and presented severe diarrhea and recurrent infections. He underwent hematopoietic stem cell transplantation at 17 months with complete donor engraftment and partial immunity improvement. In the second year of life, he progressively developed diffuse papular follicular keratoses on ichthyosiform skin, nail clubbing, and subungual hyperkeratosis. Histopathology showed hyperkeratosis with follicular plugging and scattered apoptotic keratinocytes, visualized at an ultrastructural examination. Our findings expand the spectrum of dermatological manifestations which can develop in MIA-CID patients. Examination of further patients will allow defining whether keratinocyte apoptosis is also a disease feature.

## 1. Introduction

Multiple intestinal atresia with combined immune deficiency (MIA-CID, OMIM #243150) is a rare and severe autosomal recessive syndrome due to mutations in the TTC7A gene [1,2,3]. TTC7A encodes for tetratricopeptide repeat domain 7A, a scaffolding protein involved in the polarization and differentiation of intestinal and thymic epithelial cells. Biallelic mutations in the TTC7A gene cause a spectrum of disorders comprising very early onset inflammatory bowel disease, multiple intestinal atresias without or with immunodeficiency of varying severity, or chronic intestinal pseudo-obstruction and lymphoproliferative syndrome [1,2,3,4]. Total parenteral nutrition dependency, recurrent infections, and intestinal complications lead to an early demise in the majority of MIA-CID patients [1,2,3]. In a few cases, ichthyosis-like skin lesions and variable adnexal involvement have been reported [5,6,7,8].

We describe the peculiar skin phenotype of a child affected with MIA-CID due to a previously undescribed TTC7A homozygous mutation.

## 2. Materials and Methods

Histopathological and ultrastructural analyses. Skin shaving biopsies were obtained after parents’ informed consent and processed for histopathological and ultrastructural examination, according to standard methods.

Molecular genetic diagnosis. Following informed consent, genomic DNA of the patient and his parents was extracted from peripheral blood using QIAsymphony DSP DNA Mini Kit (Qiagen, Hilden, Germany). Genetic diagnosis was obtained by TTC7A gene testing using direct Sanger sequencing of PCR fragments covering the entire coding region and flanking intron/exon junctions (NM_020458.4 reference sequence). Pathogenicity of the identified variant was evaluated by computational analysis using SIFT, PolyPhen, MutationTaster, Provean, NNSPLICE (0.9 version), Human Splicing Finder (version 3.1), and MaxEntScan algorithms.

## 3. Results

### 3.1. Case Report

A 1-year-old male was referred to our Pediatric Oncohematology Department. He was born to healthy non-consanguineous parents by vaginal delivery at 36 weeks of gestation, which had been complicated by polyhydramnios. Surgery performed in the first days of life for bowel occlusion revealed pyloric and extensive colon atresia. The neonate underwent pyloroplasty, partial colectomy, and resection of the ileocecal valve. He was on total parenteral nutrition (TPN) since birth. In the first months of life, he developed recurrent bacterial, viral, and mycotic infections suggestive of immunodeficiency, which was confirmed by laboratory findings of lymphopenia (total lymphocytes 700 cells/µL, CD3+ T cells 165 cells/µL, and memory B cells 34 cells/µL), severe hypogammaglobulinemia (IgG 76 mg/dL, IgA 12 mg/dL; IgM < 4 mg/dL), impaired mitogen response (CD3 and PHA), and low T-cell receptor excision circles (TRECs) (0.6/µL, n.v. > 25/µL). Due to persistent diarrhea, recurrent infections, failure to thrive, and dependency on TPN, sequential hematopoietic stem cells (HSC) and intestinal transplantation were planned. Following a conditioning regimen with treosulfan (42 g/m^2^), fludarabine (160 mg/m^2^), and ATG (12 mg/kg), the infant underwent HSC transplantation (HSCT) from the HLA-haploidentical mother at 17 months of age. Donor engraftment was complete (100% donor chimerism). However, diarrhea and recurrent infection persisted, and the patient developed a progressive liver disease with portal hypertension.

During hospitalization, asymptomatic skin papules appeared on the face and trunk (Figure 1A). Before the onset of cutaneous lesions, total lymphocyte count was 290/µL, with CD3+ T cells 103 cells/µL (CD4+ 27/µL, CD8+ 45/µL) and CD19+ B cells 2 cells/µL. Serum IgA and IgM immunoglobulin levels were 16 mg/dL and 31 mg/dL, respectively (the patient received post-HSCT IgG replacement therapy). Physical examination showed numerous whitish, keratotic follicular papules on diffusely xerotic and thin skin. Nail clubbing with mild distal subungual hyperkeratosis was also present (Figure 1B), while terminal hair and teeth were not affected. The follicular lesions gradually increased in number, and one erythematous nodule surmounted by a thick scale appeared on the right thigh at the age of 2 years (Figure 1C). Dermoscopy of a trunk papule showed follicular plugging surrounded by diffuse fine whitish scaling (Figure 1D). Due to the lack of symptoms and the patient’s general condition, only emollients were applied. The child was discharged at the age of 3.4 years with TPN. He then underwent liver-intestine transplantation from the mother in another hospital and died shortly afterward of transplant-related complications.

### 3.2. Histopathological and Ultrastructural Findings

Shaving biopsies of the thigh nodule and one follicular papule were performed. Histopathological examination documented modest epidermal hyperplasia, massive compact hyperkeratosis with focal areas of parakeratosis, and hypogranulosis (Figure 2A,B). Follicular ostia were dilated and filled with keratin plugs surrounding dystrophic vellus hair shafts (Figure 2B). Finally, scattered apoptotic keratinocytes with a pyknotic nucleus and an eosinophilic cytoplasm were observed in the upper epidermal cell layers (Figure 2C). At the ultrastructural examination, the stratum corneum appeared thickened, and its 3 to 4 innermost cell layers showed keratin tonofilaments embedded in a clear matrix, compatible with transitional cells. In the granular layer, keratohyalin granules appeared reduced in number and size. Finally, isolated apoptotic keratinocytes with a shrunken nucleus, nuclear chromatin condensation, and a clear cytoplasm almost devoid of organelles were observed in the suprabasal cell layers (Figure 2D).

### 3.3. Mutation Analysis

Taken together, the clinical and laboratory findings led to the diagnosis of MIA-CID and prompted us to screen for mutations in the causative gene TTC7A. Direct sequencing of all exons and flanking intronic regions revealed the missense variant c.295A>G; p.M99V in TTC7A exon 2 (NM_020458.4) (Figure 3A) at homozygous and heterozygous state in the affected proband and his healthy parents, respectively. This variant has not been previously published in scientific literature. It is extremely rare (rs74806891) in the human population, being present in the GnomAD database with an allele frequency of 0.0004% (1 allele count: 251,481 allele numbers). All four pathogenicity predictors addressed consistently classified the p.M99V substitution as not harmful. In contrast, in silico splice site finders (NNSPLICE, Human Splicing Finder, MaxEntScan) reported at the site of the c.295A>G transition, the creation of a donor splice site (AAGATGAGC AAGGTGAGC) with score values which essentially matched that of the downstream natural site (0.98 vs. 1.0, according to NNSPLICE prediction). The newly formed donor site is, thus, expected to compete against the natural one, causing aberrant splicing of a 54-nucleotide shortened exon 2, which translates a TTC7A protein internally deleted of 18 amino acids (p.M99_S116del) (Figure 3B).

## 4. Discussion

Just over 50 patients with TTC7A mutations have been reported since the original description in 2013 [1,2,3,4,9,10,11]. We considered the homozygous c.295A>G variant as disease-causing since patient phenotype and laboratory findings are diagnostic for MIA-CID. This variant has not been previously reported in the literature and Human Gene Mutation Database (http://www.hgmd.cf.ac.uk, accessed on 10 March 2022). Furthermore, allele frequency (0.0004%), disease rarity, and mode of inheritance (homozygosity) support its role in causing disease. Finally, computational analysis with software based on different principles consistently assigns to it the valence for inducing aberrant splicing. More than half of MIA-CID patients die within the first year of life, mainly due to bowel obstruction, sepsis, and other infections. Genotype-phenotype comparisons indicated that patients with biallelic homozygous or compound heterozygous missense mutations not affecting the tetratricopeptide repeat (TPR) domains usually have a more favorable prognosis [9,10]. Indeed, in our patient, the in-frame deletion p.M99_S116del, predicted from the aberrant splicing of TTC7A exon 2, falls upstream of the first N-terminal TPR domain (which encompasses residues 121 to 157) and does not affect the remaining TPR2-9 functional domains. Alternatively, it can be hypothesized that the c.295A>G transition produces leaky effects on splicing, allowing the synthesis of scant amounts of full-length transcripts and wild-type protein.

Our patient was alive at 17 months of age when he successfully underwent HSCT. HSCT has been reported to improve disease course and outcome, without efficacy on gastrointestinal manifestations [12]. HSCT allowed partial immunity improvement with a decrease in recurrent infections in our case. However, bowel disease persisted, requiring specific medical treatment and continuous TPN, which in turn was likely implicated in progressive liver disease. A single patient affected with combined MIA-CID phenotype and liver cirrhosis, without identified genetic defects, successfully underwent liver-small intestine transplantation with recovery of liver function, intestinal motility, and enteral feeding [13]. Indeed, our patient eventually underwent liver-intestine grafting but unfortunately died shortly thereafter due to transplant-related complications.

In the second year of life, the child developed a skin and adnexal phenotype consisting of prominent follicular keratoses, subungual hyperkeratosis, and mild diffuse ichthyosis-like scaling. Previously reported skin manifestations in MIA-CID and in a closely-related allelic condition named enteropathy-lymphocytopenia-alopecia syndrome are summarized in Table 1 [5,6,7,8]. They comprise scarring alopecia, psoriasis-like lesions, dry skin with progressive ichthyosis, excoriated and eczematous lesions, palmoplantar keratoderma (PPK), and subungual hyperkeratosis. In particular, one MIA-CID patient presented skin dryness at 3 years, and a second xerosis associated with PPK at age 6. In these patients, histopathological examination of skin biopsies showed an ichthyotic epidermis and follicular plugging [7]. Ichthyosis-like histopathological features were also present in another patient [8]. Indeed, flaky skin mice carrying a spontaneous mutation in the Ttc7 gene present skin anomalies, including patchy alopecia and skin scaling [7,14]. The mouse phenotype confirms that dermatologic features in MIA-CID patients are disease-specific. The development of cutaneous and adnexal lesions in early childhood, together with disease lethality in the first year of life in the majority of patients, could partly explain the description of dermatologic findings in a few cases to date. Interestingly, our patient manifested prominent and diffuse follicular hyperkeratosis documented by dermoscopy and histopathology. Although follicular keratoses have not been previously reported, the histological finding of dilated follicular ostia filled with keratin plugs has been described in two patients who underwent skin biopsy [7]. It has been suggested that epidermal alterations may reflect an altered equilibrium between epidermal maturation and differentiation in TTC7A-deficient keratinocytes [7]. Abnormal epidermal maturation and differentiation might also affect follicular infundibula resulting in clinically evident follicular keratosis, at least in some patients. An additional finding in our patient was the detection of apoptotic keratinocytes by both histopathological and ultrastructural examination. Indeed, apoptotic enterocytes are well-recognized MIA-CID disease markers [3,5,6]. Apoptotic keratinocytes seen in our patient suggest that common pathogenetic mechanisms may underlie both skin and bowel manifestations. However, since lesion shaving was performed after HSCT, we cannot exclude that apoptotic cells represent the histopathological counterpart of subclinical graft-versus-host disease. Clinical, histopathological, and molecular genetic studies of further patients will allow us to better define the features and frequency of dermatological manifestations in patients carrying TTC7A mutations, to clarify their pathogenesis, and to determine if keratinocyte apoptosis can be part of the MIA-CID spectrum.

## Figures and Tables

**Figure 1 genes-13-00821-f001:**
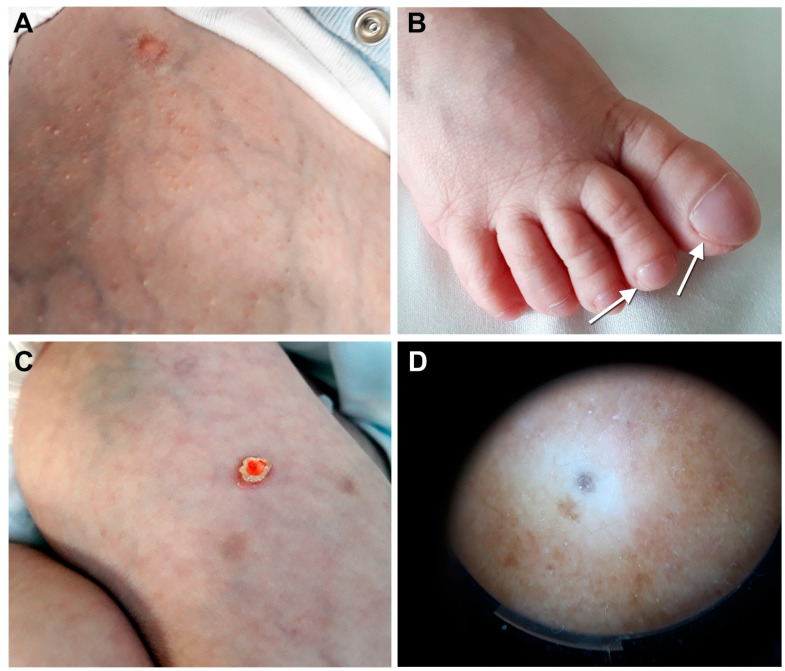
Patient’s clinical features at the age of 2 years. (**A**) Multiple whitish follicular papules on the abdomen, note the thin skin with prominent superficial veins related to liver disease and portal hypertension; (**B**) mild subungual hyperkeratosis (arrows) with nail clubbing and accentuated skin markings on the foot dorsum; (**C**) a single erythematous nodule surmounted by a thick scale is visible on the right thigh; the central red color is due to an antiseptic solution; (**D**) dermoscopy of a trunk papule shows follicular plugging surrounded by diffuse fine whitish scales.

**Figure 2 genes-13-00821-f002:**
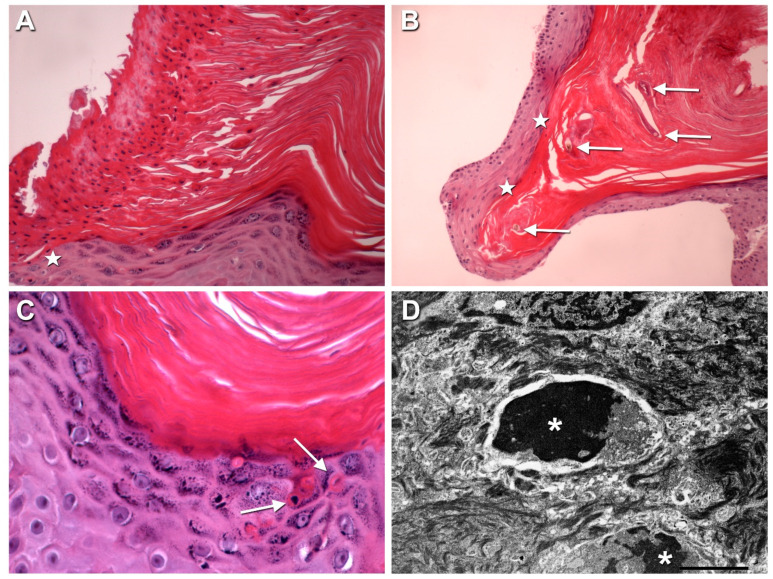
Histopathological and ultrastructural findings. (**A**,**B**) Histopathological examination of the thigh nodule shaving biopsy shows compact massive hyperkeratosis with a column of parakeratotic cells (**A**), and a keratin plug surrounding a dystrophic vellus hair shaft (**B**, arrows); note the focal hypogranulosis below the parakeratotic column (star in **A**) and in the epidermis around the keratin plug (stars in **B**). (**C**) scattered apoptotic keratinocytes with a pyknotic nucleus and an eosinophilic cytoplasm (arrows) are visible in the upper epidermal cell layers. (**D**) Ultrastructural examination of a follicular papule shows two apoptotic keratinocytes with shrunken condensed nuclei (asterisks) (bar: 2 µm). (**A**–**C**): hematoxylin-eosin staining, original magnification 200× in (**A**), 100× in (**B**), 400× in (**C**).

**Figure 3 genes-13-00821-f003:**
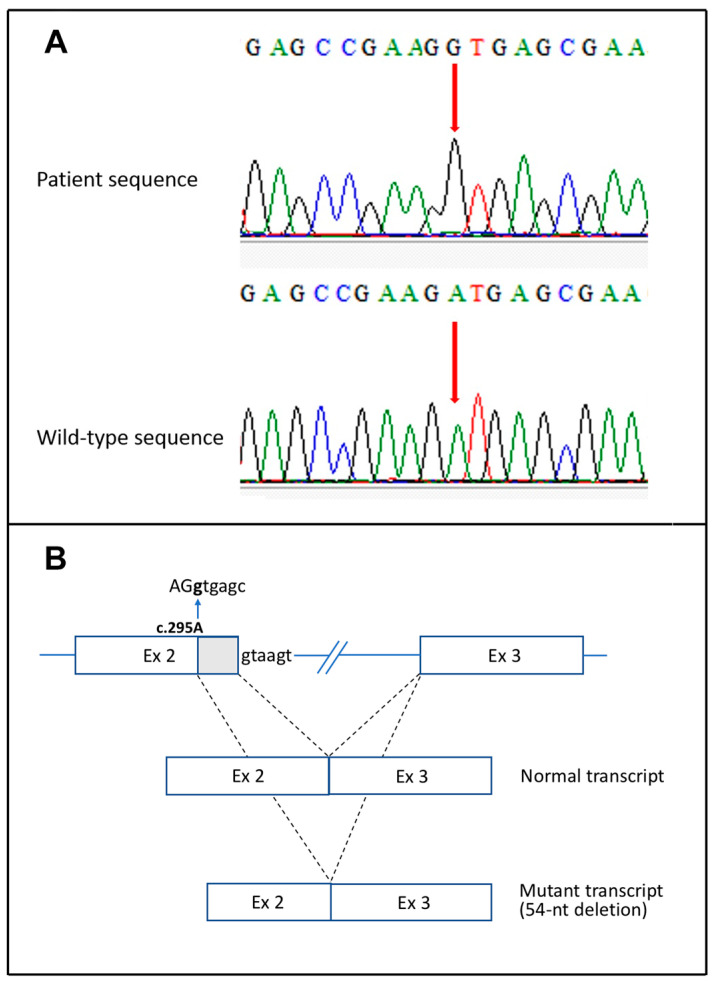
(**A**) Electropherograms of the direct sequence of exon 2 TTC7A gene: the homozygous missense mutation c.295A>G (p.M99V) is shown (red arrows). (**B**) Potentially altered splicing due to the c.295A>G mutation. The mutation creates a new donor splice site (AGgtgagc) in a late exonic position that results in an aberrant exon 2 definition. Usage of this cryptic donor site leads to a mutant in-frame transcript shortened of 54 nucleotides (p.M99_S116del). Normal splicing by the naturally occurring donor site (gtaagt) is shown for comparison.

**Table 1 genes-13-00821-t001:** Dermatological, histopathological, and molecular features of patients carrying TTC7A mutations.

Patient n.^ [Ref.° n.]	Age, Gender	TTC7A Mutation(s) (c.DNA, Protein)	Dermatological Features	Histopathology
1 [6,7]	3 years, female	c.1008C>G, p.Tyr336*; c.1479delG, p.Leu493fs*13	Diffuse xerosis	Ichthyotic epidermis, epidermal hyperplasia, follicular keratin plugs
2 [6,7]	6 years, male	c.2496_2497delCG, p.A832fs*1	Diffuse xerosis, PPK”, acral and subungual hyperkeratosis	Ichthyotic epidermis, epidermal hyperplasia, follicular keratin plugs
3–7 (single-family) [5,7]	4–50 years, 3 females, 2 males	c.211G>A, p.E71K	Progressive alopecia (all cases), toenail subungual hyperkeratosis (4 cases), psoriasiform lesions (1 case)	Not available
8 [8]	5 years, female	c.2170C>A, p.Q724K	Diffuse xerosis, eczematous lesions, and excoriations	Hyperorthokeratosis, spongiosis, eosinophilic infiltrate
Present case	20 months, male	c.295A>G, p.M99V	Diffuse xerosis, follicular hyperkeratosis, mild subungual hyperkeratosis	Epidermal hyperplasia, follicular keratin plugs, apoptotic keratinocytes

^n.: number; °: reference; ”: Palmoplantar keratoderma.

## Data Availability

The data presented in this study are available on request from the corresponding author. The data are not publicly available due to privacy reasons.

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
