# Peer review of "Prominent Follicular Keratosis in Multiple Intestinal Atresia with Combined Immune Deficiency Caused by a TTC7A Homozygous Mutation"

_genes, 2022, doi:10.3390/genes13050821_

Round 1

Reviewer 1 Report

I appreciate your invitation to revise manuscript number " genes-1667598". Overview: the current case report described the association between Prominent Follicular Keratosis and Multiple Intestinal Atresia 2 with Combined Immune Deficiency Caused by a TTC7A Ho-3 mozygous Mutation

The description of case seems to be good. I have just a few questions/suggestions.

  • As this case somewhat lacks novelty, I suggest that the authors should give more details and explanations describing the relation between Follicular Keratosis and TTC7A Ho-3 mozygous Mutation (in the discussion section).
  • What about the treatment of clinical manifestations in this case.
  • The authors should give more details about any Immunologic parameters measured before the onset of skin lesions.

Author Response

Comment. The description of case seems to be good. I have just a few questions/suggestions. As this case somewhat lacks novelty, I suggest that the authors should give more details and explanations describing the relation between Follicular Keratosis and TTC7A homozygous Mutation (in the discussion section).

Answer. We thank the Reviewer for his/her comments and suggestions aimed at improving the quality of our manuscript. Specific to the first comment, although clinically evident follicular keratoses have not been reported to date in patients with TTC7A mutations, the histological finding of dilated follicular ostia filled with keratin plugs has been described in the skin biopsy of two patients (Leclerc-Mercieret al. Br. J. Dermatol. 2016, 175, 1061-64, ref. #7 in the manuscript). The Authors suggested that epidermal alterations may reflect an altered equilibrium between epidermal maturation and differentiation in TTC7A-mutated keratinocytes. Abnormal epidermal maturation and differentiation might also affect follicular infundibula resulting in clinically evident follicular keratosis, at least in some patients. We have added two sentences detailing this aspect in the Discussion section (page 7, lines 194-201). In addition, we have modified the last sentence of the Discussion to point out that further studies are needed to completely describe the spectrum of dermatological manifestations in patients carrying TTC7A mutations and to clarify their pathogenesis.

Comment. What about the treatment of clinical manifestations in this case.

Answer. Due to the lack of symptoms and patient’s poor general condition, we decided to apply emollients only, as now specified in the Results section at page 2, Case report paragraph, lines 92-93.

Comment. The authors should give more details about any Immunologic parameters measured before the onset of skin lesions.

Answer. As requested, immunologic parameters before the onset of skin lesions have been added to the Results section, Case report paragraph, page 2, lines 81-88

Reviewer 2 Report

The case described is interesting and new in the literature. The case is well described, provides sufficient information and is properly discussed. In the opinion of this reviewer, it has the necessary elements to be published in this form.

Author Response

Comment. The case described is interesting and new in the literature. The case is well described, provides sufficient information and is properly discussed. In the opinion of this reviewer, it has the necessary elements to be published in this form.

Answer. We thank the Reviewer for his/her consideration of our manuscript and kind comments.